# A Description and Safety Overview of Irreversible Electroporation for Prostate Tissue Ablation in Intermediate-Risk Prostate Cancer Patients: Preliminary Results from the PRESERVE Trial

**DOI:** 10.3390/cancers16122178

**Published:** 2024-06-08

**Authors:** Arvin K. George, Ranko Miocinovic, Amit R. Patel, Derek J. Lomas, Andres F. Correa, David Y. T. Chen, Ardeshir R. Rastinehad, Michael J. Schwartz, Edward M. Uchio, Abhinav Sidana, Brian T. Helfand, Jeffrey C. Gahan, Alice Yu, Srinivas Vourganti, Al Baha Barqawi, Wayne G. Brisbane, James S. Wysock, Thomas J. Polascik, Timothy D. McClure, Jonathan A. Coleman

**Affiliations:** 1VA Ann Arbor Health System, Ann Arbor, MI 48105, USA; 2Michigan Medicine, Ann Arbor, MI 48109, USA; 3Johns Hopkins University, Brady Urological Institute, Baltimore, MD 21287, USA; 4Duly Health and Care, Downers Grove, IL 60515, USA; 5Mayo Clinic, Rochester, MN 55905, USA; 6Fox Chase Cancer Center, Philadelphia, PA 19111, USA; 7Northwell Health System, Manhasset, NY 11030, USA; 8University of California, Irvine, Irvine, CA 92697, USA; 9University of Cincinnati, College of Medicine, Cincinnati, OH 45221, USA; 10University of Chicago, Section of Urology, Chicago, IL 60637, USA; 11Northshore University HealthSystem, Northshore University HealthSystem Research Institute, Evanston, IL 60201, USA; 12University of Texas, University of Texas Southwestern Medical Center, Dallas, TX 75390, USA; 13Moffitt Cancer Center, Tampa, FL 33612, USA; 14Rush University, Rush University Medical Center, Chicago, IL 60612, USA; 15University of Colorado, University of Colorado Anschutz Medical Campus, Aurora, CO 80045, USA; 16University of Florida, Gainesville, Gainesville, FL 32611, USA; 17NYU Langone Health, NYU Grossman School of Medicine, New York, NY 10016, USA; 18Duke University, Duke University School of Medicine, Durham, NC 27705, USA; 19Cornell University, Weill Medical College, New York, NY 10065, USA; 20Memorial Sloan Kettering Cancer Center, New York, NY 10065, USA

**Keywords:** prostate cancer, irreversible electroporation, ablation, focal therapy, clinical trial

## Abstract

**Simple Summary:**

The PRESERVE study is the first, large, prospective, pivotal trial of irreversible electroporation using the NanoKnife System for the prostate in the United States. The data from this United States Food and Drug Administration Investigational Device Exemption study aim to evaluate the safety and effectiveness of irreversible electroporation with the NanoKnife System to ablate prostate tissue in patients with intermediate-risk prostate cancer.

**Abstract:**

The PRESERVE study (NCT04972097) aims to evaluate the safety and effectiveness of the NanoKnife System to ablate prostate tissue in patients with intermediate-risk prostate cancer (PCa). The NanoKnife uses irreversible electroporation (IRE) to deliver high-voltage electrical pulses to change the permeability of cell membranes, leading to cell death. A total of 121 subjects with organ-confined PCa ≤ T2c, prostate-specific antigens (PSAs) ≤ 15 ng/mL, and a Gleason score of 3 + 4 or 4 + 3 underwent focal ablation of the index lesion. The primary endpoints included negative in-field biopsy and adverse event incidence, type, and severity through 12 months. At the time of analysis, the trial had completed accrual with preliminary follow-up available. Demographics, disease characteristics, procedural details, PSA responses, and adverse events (AEs) are presented. The median (IQR) age at screening was 67.0 (61.0–72.0) years and Gleason distribution 3 + 4 (80.2%) and 4 + 3 (19.8%). At 6 months, all patients with available data (n = 74) experienced a median (IQR) percent reduction in PSA of 67.6% (52.3–82.2%). Only ten subjects (8.3%) experienced a Grade 3 adverse event; five were procedure-related. No Grade ≥ 4 AEs were reported. This study supports prior findings that IRE prostate ablation with the NanoKnife System can be performed safely. Final results are required to fully assess oncological, functional, and safety outcomes.

## 1. Introduction

Prostate cancer (PCa) is the second most common cancer and the fifth leading cause of cancer death in the United States. Due to advances in diagnostic testing, approximately 70% of PCa cases are diagnosed when localized and have a five-year survival rate of >99% [1,2,3]. Patients with very low- and low-risk PCa have minimal risk of metastatic disease and mortality [4], and guidelines recommend active surveillance for these patients [2]. While some patients with intermediate-risk PCa can benefit from active surveillance, for others, the risk of metastasis may warrant treatment [5], with radical prostatectomy and radiation therapy considered the standard of care for this patient population [2,3]. Despite effective cancer control, radical treatments can have negative effects on sexual, urinary, and bowel functions [6,7]. Furthermore, an analysis from the ProtecT study concluded that while radical treatments reduced the incidence of metastasis, there were no differences in PCa-related mortality at 15 years among all risk groups [8].

Focal therapy is an alternative treatment strategy that can delay or eliminate the need for whole-gland treatment by targeting only areas of the prostate with clinically significant disease. Functional benefits of focal therapy are well documented, including significantly less incontinence or erectile dysfunction than observed with radical treatment [9].

Irreversible electroporation (IRE) is a promising method of focal ablation that uses high-voltage electrical pulses to change the permeability of the cell membrane, leading to cell death [10,11]. A potential benefit of IRE is that it is largely athermal [12,13], which can avoid perfusion-associated sink effects with thermal-based therapies such as high-intensity focused ultrasound ablation or cryotherapy. The PRESERVE trial (ClinicalTrials.gov NCT04972097) is a United States Food and Drug Administration (FDA) Investigational Device Exemption (IDE) study which aims to evaluate the safety and effectiveness of IRE with the NanoKnife System to ablate prostate tissue in patients with intermediate-risk PCa.

## 2. Materials and Methods

The PRESERVE study is a prospective, non-randomized, pivotal trial of subjects in the United States from 17 clinical centers sponsored through the Society of Urologic Oncology Clinical Trials Consortium. Enrollment began in March 2022 and ended in July 2023 with follow-up through August 2024.

Multiparametric magnetic resonance imaging (mpMRI) was completed within 180 days prior to enrollment to identify the presence of suspicious lesions using the Prostate Imaging Reporting and Data System (PIRADS) v2.1 [14]. On screening mpMRI, the absence of gross extraprostatic extension and seminal vesicle invasion was required. The patients’ prostate lesions were sampled by ultrasound-guided transperineal or transrectal prostate biopsy within 180 days prior to study enrollment to confirm the presence of cancer.

Eligible subjects were men greater than 50 years of age with organ-confined PCa that was clinical stage ≤ T2c. Subjects were required to have prostate-specific antigens (PSAs) ≤ 15 ng/mL or a PSA density < 0.15 ng/mL2 if PSAs > 15 ng/mL and Gleason score of 3 + 4 or 4 + 3. Among the exclusion criteria were active urinary tract infection, any prior or current PCa therapy, and in the opinion of the treating physician, any contraindication for treatment (unable to undergo MRI, etc.). Complete eligibility criteria are provided in Appendix A.

At the time of analysis, post-treatment early follow-up on all patients was available. Subjects will be followed for 12 months after receiving IRE with follow-up encounters occurring 3–10 days post-procedure and 1, 3, 6, 9, and 12 months post-procedure. Adverse events, PSAs, and functional outcomes are reported at each follow-up visit. Functional outcomes are assessed using validated instruments including the UCLA Expanded Prostate Cancer Index Composite Urinary domain, International Prostate Symptom Score (IPSS), and IPSS Quality of Life scores, the International Index of Erectile Function-15, and the EuroQoL. Post-treatment imaging will be completed using mpMRI with contrast at 3 and 12 months post-procedure. Fusion biopsy will be completed at the 12-month follow-up visit and earlier if required “for cause” at the clinician’s discretion. The complete follow-up schema is presented in Appendix A.

All subjects provided written informed consent. The study protocol was approved by local or central institutional review boards depending on the center. The study was conducted in accordance with the International Conference on Harmonization’s Good Clinical Practice guidelines.

The MRI-visible and histopathologically confirmed area was treated with IRE using the NanoKnife System (AngioDynamics, Inc., Latham, NY, USA), see Appendix A. A treatment margin of ≥5 mm was applied to the MR-visible lesion.

Subjects were placed in dorsal lithotomy under general anesthesia with endotracheal intubation, and the procedures were completed under sterile technique. A 16 Fr Foley catheter was placed for urethral landmarking and remained for several days post-operatively for bladder drainage. Biplane transrectal ultrasound guidance via a stepper unit was utilized for imaging with an 18-gauge brachytherapy grid. Commonly, a stand-off rectal balloon was placed on the ultrasound transducer to improve imaging quality. Sagittal and transverse views were assessed to ensure the entire length of the prostate was visualized and the ultrasound transducer was placed midline to the prostate gland. Careful consideration of transducer opposition to the gland is important for ideal imaging quality and safety of electrode placement to the rectum. Assessment of the prostate gland and critical structure morphology is important for the planning of safe electrode placement. 

To complete the procedure, the NanoKnife System utilizes two to six 18-gauge electrodes to administer voltage at 1500 V/cm, with a pulse length of 90 µs. Most often, four to six electrodes are utilized for prostate ablation. The ideal current per pulse is ≥20 amps with a maximum of 50 amps. Interelectrode distances should be 10 to 22 mm. Electrodes are inserted via biplane ultrasound guidance to bracket the intended area of ablation. All electrode tips are positioned within prostatic tissue, 3 mm from the capsule and 5 mm away from critical structures such as the rectum. After placement of the NanoKnife electrodes in the prostate and immediately prior to energy delivery, a nondepolarizing neuromuscular blocking agent is administered to reduce skeletal muscle contraction associated with the administration of high-voltage pulses. It is important that the patient is deeply paralyzed during energy delivery (zero twitches from a train of four). Ten pulses are then administered to each electrode pairing to assess the conductance of the tissue. The amperage should begin within the range of 20–35 amps. If the amperage is <20 amps or >35 amps, adjustments to the voltage may be applied to compensate. After ideal conductance is obtained between all electrode pairs, 80 remaining pulses are delivered. Active energy delivery time ranges from 9 to 15 min depending on the number of electrode pairings utilized.

Once energy delivery was completed, the electrodes were removed and a gauze dressing with antimicrobial ointment was applied to the perineum. Patients were discharged from the hospital or ambulatory surgery center within 24 h of the procedure.

The primary objectives of this study were to evaluate the effectiveness of ablation by the negative in-field biopsy rate at 12 months and procedural and post-treatment safety via the incidence and severity of adverse events (AEs) through 12 months. Secondary objectives included the assessment of urinary and erectile function after IRE using validated questionnaires (IIEF-15, I-PSS, EPIC-U, and EQ-5D-5L), PSA kinetics, and the need for secondary or adjuvant treatments. Preliminary results on patient demographic and disease characteristics, procedure details, PSA response, and adverse events are presented.

AEs were coded according to the Medical Dictionary for Regulatory Activities (MedDRA) version 24.1 and graded by Common Terminology Criteria for Adverse Events (CTCAE) version 5.0 severity with severe AEs defined as Grade 3 or higher.

Statistical analysis was performed using SAS version 9.4. Results are presented on the intent-to-treat population which was defined as all subjects who were enrolled and treated. Data for continuous variables are presented using descriptive statistics including the median, interquartile range (IQR), minimum, and maximum. Categorical variables are presented using descriptive statistics including frequencies and percents. Time to PSA nadir was obtained via Kaplan–Meier estimation and subjects who did not encounter the event were censored at the date of last observation. Missing PSA data were imputed using the last observation carried forward.

## 3. Results

### 3.1. Study Population and Disease Characteristics

Among 151 subjects screened, 128 were enrolled and 121 received IRE treatment across 17 centers. At the time of analysis, 87.6% of subjects completed their 3-month visit, 62.0% completed their 6-month visit, and 17.4% completed their 12-month visit, with a median (IQR) follow-up of 6.1 (3.1–9.1) months. Median (IQR) age at informed consent was 67.0 (61.0–72.0) years. The American Joint Committee on Cancer (AJCC) primary tumor stage was available for 119 subjects, and 86.6% were stage T1c. Gleason score distribution at screening was 3 + 4 (80.2%) and 4 + 3 (19.8%) (see Appendix A). Note that there was no central pathology lab; biopsy data were reported per local site pathology reads.

Lesion location was recorded for 116 subjects. Among these, 41.4% of treated tumors were located in the apex, 14.7% were located in the base, and 44.0% were mid-gland. The median (IQR) lesion diameter was 10.0 (6.0–15.0) mm. One lesion was treated per patient. Tumor location was anterior for 43.1% of subjects and posterior for 56.9%. The median (IQR) procedure duration was 50.0 (38.0–63.5) min and ranged from 18 to 128 min. The median (IQR) number of probes used was 4 (4–5). A majority (86.8%) used an ultrasound/MRI fusion device during the procedure. One subject was re-treated with IRE for a new out-of-field significant lesion. A Foley catheter was placed in 97.5% of subjects and these subjects were catheterized for a median (IQR) of 3.0 (3.0–5.0) days (Table 1).

### 3.2. Biochemical Response

The median (IQR) PSA was 5.8 (4.8–7.7) ng/mL at screening. By 6 months post-procedure, all patients with data through that timepoint (n = 74) had experienced a reduction in PSA. The median (IQR) decreases in PSA measured at 3 and 6 months post-procedure were 3.7 (2.2–5.7) ng/mL and 3.5 (2.4–5.4) ng/mL, respectively. The median (IQR) percent decreases at 3 and 6 months post-procedure were 67.2% (46.6–83.2%) and 67.6% (52.3–82.2%), respectively. The median (IQR) PSA dropped from 5.8 (4.8–7.7) ng/mL at baseline to 1.8 (1.0–2.9) ng/mL at 3 months and 1.6 (1.1–3.0) ng/mL at 6 months (Figure 1). The median (IQR) PSA nadir was 1.6 (0.9–2.7) ng/mL, and the median (IQR) time to PSA nadir was 3.0 (1.2–5.8) months.

### 3.3. Safety Overview

The majority of AEs (96%) were CTCAE Grade 1 or Grade 2. Ten subjects (8.3%) experienced a Grade 3 AE (i.e., severe AE), with a total of 13 events. No Grade 4 or 5 AEs were reported; there were no deaths. Grade 3 AEs by system organ class and preferred term are presented in Table 2.

Regarding AEs related to IRE, 74 (61.2%) subjects experienced any AE related to IRE and five (4.1%) subjects experienced a Grade 3 AE related to IRE. These five subjects each experienced one Grade 3 AE (Table 3): urinary retention (n = 2), bladder spasm (n = 1), abdominal pain (n = 1), and rectourethral fistula (n = 1). The abdominal pain began the day after treatment and resolved the next day and was accompanied by nausea and intermittent vomiting. It was a self-limited episode and resolved spontaneously, presumably secondary to anesthesia. The fistula occurred three months post-treatment and resolved after conservative management with prolonged suprapubic catheter placement.

The most common adverse events occurring in ≥10% of subjects included hematuria (38.0%), dysuria (13.2%), urinary retention (11.6%), micturition urgency (11.6%), pollakiuria (9.9%), erectile dysfunction (9.9%), and hematospermia (9.9%) (Table 4). The majority of adverse events occurred in the first 30 days post-procedure (Table 5) and resolved without intervention.

## 4. Discussion

The PRESERVE study is an FDA IDE study and the first, large, United States-based trial evaluating IRE for the ablation of prostate tissue in subjects with intermediate-risk PCa. Subjects experienced a low number of severe AEs and no AEs resulting in life-threatening consequences or death. This study further supports evidence from prior studies regarding the safety of IRE for the treatment of localized PCa [15,16,17,18]. As with prior studies, subjects largely experienced CTCAE Grade 1 and 2 AEs, a majority of which were hematuria and dysuria and were anticipated. Only five subjects experienced Grade 3 AEs related to the IRE procedure (urinary retention, bladder spasm, abdominal pain, and rectourethral fistula). The low number of severe AEs impacting the bladder is a promising finding since IRE aims to minimize urinary dysfunction. Similarly, AEs that could potentially impact sexual function were low. Only one severe AE impacted the reproductive system (testicular pain), which was not related to the procedure.

As noted above, one patient developed a delayed rectoprostatic urethra fistula, which presented at 3 months post-IRE. The subject received an IRE ablation for an 18 mm lesion located in the posterior peripheral zone; hydrodissection was not used. The posterior aspect of the gland was treated from the apex to the base. The clinical presentation included watery diarrhea while voiding very small amounts for the six weeks prior. The treating physician attributed it to a robotic inguinal hernia repair surgery and urethral instrumentation, which occurred 4 weeks after IRE (2 weeks before the development of symptoms), resulting in delayed presentation. The patient underwent cystoscopy, demonstrating a large prostatic fossa, high bladder neck, and small fistulous opening. Conservative management with a suprapubic catheter was elected. After an initial complete resolution of symptoms, they recurred after suprapubic catheter removal at 8 weeks. The suprapubic catheter was replaced for an additional 12 weeks. The fistula resolved after demonstrating normal voiding and bowel function with the suprapubic catheter clamps for the latter 6 weeks of placement.

Zhang et al. summarized safety outcomes for a cohort of 411 patients treated in a multi-center prospective registry [18]. They reported a low rate of adverse events—2.6% overall—with the majority being Grade 1 and Grade 2.

A recent meta-analysis by Ong et al. focusing on outcomes from nine publications reported minimal severe adverse events [19]. Grade 1 and 2 complications of hematuria, dysuria, urinary tract infection, pain, urgency, and temporary incontinence were most common, which support the findings of the current study.

In a systematic review of 14 studies and nearly 900 patients treated with IRE, Prabhakar et al. reported that the most common complications associated with IRE were urinary retention (5.6–26.3%), hematuria (6.7–24%), dysuria (15–26.3%), and urinary tract infection (9–11%), predominantly Grade 1 and Grade 2 [20]. The authors noted a 0.2% rate of rectourethral fistula reported in one study, confirming this complication to be rare as seen in the current trial. Three Grade 3 events were reported across all studies reviewed, including non-ST elevation myocardial infarction, epididymitis leading to abscess formation, and urethral stricture requiring urethrotomy. The safety profile of IRE was found to be similar to that of other focal therapy modalities, such as HIFU, cryotherapy, photodynamic therapy, and focal laser ablation [21,22,23].

Valerio et al. reported the initial safety and feasibility of 34 patients undergoing IRE, demonstrating at Grades 1–2 an AE rate of 64.7%, with no Grade ≥ 3 AEs [24]. Similar to the current study, hematuria (15%) and dysuria (15%) were the most common, with a urinary tract infection rate of 15% and only 6% of patients experiencing urinary retention.

With respect to PSA, Collettini et al. reported on a prospective study of 30 patients treated with IRE, who saw a median percent reduction in PSA of 69% at 6 months. Similarly, van den Bos et al. reported a median PSA reduction of 70% between the baseline and 6–12 months post-operatively for a prospective cohort of 63 patients undergoing IRE [17,25]. These results confirm the reduction in PSA seen in the current study.

Tumor location in this study was anterior for 43% and posterior for 57% of subjects. When determining which focal therapy modality to use for the treatment of PCa, studies have proposed selecting the modality based on lesion localization and other tumor and patient characteristics [26,27]. A potential benefit of IRE similar to other modalities that utilize a transperineal approach is that it can be used safely regardless of tumor location. Scheltema et al. demonstrated that focal IRE can be performed safely on all prostate segments with no statistically significant differences between segments in terms of functional and quality of life outcomes [28]. Another potential benefit of IRE is that, due to its largely athermal mechanism of action, it may have less risk of damage to sensitive surrounding structures such as neurovascular tissue [29]. The fistula encountered in one patient in the study was associated with urethral instrumentation during hernia repair procedure shortly after the IRE procedure when tissues may have been healing and vulnerable to injury. Results from a stage IIa trial demonstrated low genitourinary toxicity and preservation of continence in men with localized PCa treated with IRE [15].

Follow-up data are still being collected for this study, which limited our ability to present results on biopsy-proven efficacy. The analyses of biochemical response revealing uniform PSAs decreasing at 3 and 6 months post-procedure is consistent with a favorable indication of ablation treatment efficacy. An analysis of 703 patients treated with focal therapy (HIFU) showed that median percent reduction in PSA was an independent predictor of the need for additional treatment and of the presence of prostate cancer at follow-up, controlling for covariates of age, baseline PSA, prostate volume, clinical stage, Gleason score, maximum cancer core length, and ablation type (quadrant- vs. hemi-ablation). This study also demonstrated that a percent PSA reduction of 50% should be considered a surrogate for good treatment effectiveness in focal therapy, leading to a decreased probability of receiving additional or radical treatment within five years [30]. However, biopsy remains the gold standard for the evaluation of oncological outcomes.

Confirmatory biopsy results and imaging will further assess oncological outcomes, including the primary outcome, rate of negative in-field biopsy at 12 months, as well as functional outcomes.

This study had several limitations. First, AE data are reported before all patients have achieved 12 months of follow-up. It is possible that patients could experience additional AEs; however, it is anticipated that many AEs, especially severe ones, and any related to treatment, would occur in the peri-operative period included in this analysis. Given that almost 90% of subjects completed their 3-month visit and delayed AEs are unlikely, conclusions regarding the safety of the procedure are unlikely to change. Second, as a single-arm trial, there is no randomized comparator and all patients received the treatment under study, and therefore only outcomes related to IRE were observed. Additionally, there was no central imaging or pathology review. Imaging and pathology interpretation are inherently subjective and there exists known variability. However, the current study represents real-world practice, as even interpretation among providers within a single clinical site can vary. Lastly, while this study included patients with intermediate-risk PCa, there is heterogeneity even within this subgroup. For example, intermediate risk could further be subdivided into favorable and unfavorable risk based on stage and/or Gleason score [31]. A majority of subjects in this study had stage T1c PCa and a Gleason of 3 + 4, which may have limited the generalizability of the results to men with intermediate-risk PCa of other severities.

## 5. Conclusions

This is the first, large, United States-based, prospective trial evaluating IRE for the ablation of prostate tissue in subjects with intermediate-risk PCa. This study supports findings from prior studies that IRE prostate ablation with the NanoKnife System can be performed safely. In our preliminary analysis of prostate tissue ablation using the NanoKnife System in patients with intermediate-risk PCa, we demonstrated that IRE can be performed safely with low toxicity and preliminary evidence of ablative efficacy in line with prior published experience. Final results from the PRESERVE trial will further assess safety, oncological, and quality of life outcomes at 12 months.

## Figures and Tables

**Figure 1 cancers-16-02178-f001:**
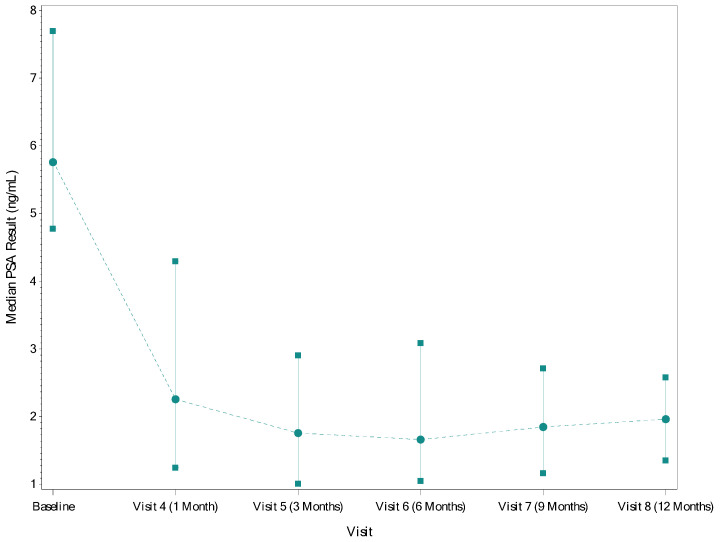
Median (IQR) PSA by Timepoint. Circles–point estimate of PSA; Squares/solid lines–IQR; Dotted lines–trend line across visits.

**Table 1 cancers-16-02178-t001:** NanoKnife System Procedure Details.

Characteristic		Study Population (N = 121)
Duration of Procedure (Minutes)	n	116
	Median	50.0
	Q1, Q3	38.0, 63.5
	Min, Max	18.0, 128.0
Number of Probes Used	n	120
	Median	4.0
	Q1, Q3	4.0, 5.0
	Min, Max	3.0, 6.0
Lesion Location Directionality Section 1	n	116
	Apex	48 (41.4)
	Base	17 (14.7)
	Midline	51 (44.0)
Lesion Location Directionality Section 2	n	116
	Anterior	50 (43.1)
	Posterior	66 (56.9)
Ultrasound/MRI Fusion Device During Procedure	n	121
	Yes	105 (86.8)
	No	16 (13.2)
Urinary Catheterization	n	121
	Yes	118 (97.5)
	No	3 (2.5)

**Table 2 cancers-16-02178-t002:** CTCAE Grade 3 Adverse Events by System Organ Class and Preferred Term.

		Study Population (N = 121)	
System Organ Class	Preferred Term	Number (%) of Subjects	Number of Events
Any Adverse Events		10 (8.3)	13
Gastrointestinal Disorders		3 (2.5)	3
	Abdominal Pain	1 (0.8)	1
	Large Intestine Polyp	1 (0.8)	1
	Rectourethral Fistula	1 (0.8)	1
Renal and Urinary Disorders		3 (2.5)	3
	Urinary Retention	2 (1.7)	2
	Bladder Spasm	1 (0.8)	1
Infections and Infestations		2 (1.7)	2
	Diverticulitis	1 (0.8)	1
	Urinary Tract Infection	1 (0.8)	1
Neoplasms Benign, Malignant, and Unspecified (incl Cysts and Polyps)		1 (0.8)	1
	Phaeochromocytoma	1 (0.8)	1
Nervous System Disorders		1 (0.8)	1
	Myasthenia Gravis	1 (0.8)	1
Reproductive System and Breast Disorders		1 (0.8)	1
	Testicular Pain	1 (0.8)	1
Respiratory, Thoracic, and Mediastinal Disorders		1 (0.8)	1
	Pulmonary Embolism	1 (0.8)	1
Vascular Disorders		1 (0.8)	1
	Deep Vein Thrombosis	1 (0.8)	1

CTCAE = Common Terminology Criteria for Adverse Events. Note: Adverse events were coded using MedDRA version 24.1. For each system organ class and preferred term, subjects are included only once in the “Number of Subjects” column, even if they experienced multiple events in that system organ class or preferred term.

**Table 3 cancers-16-02178-t003:** CTCAE Grade 3 or Greater Adverse Events Related to NanoKnife System Treatment by System Organ Class and Preferred Term.

		Study Population (N = 121)	
System Organ Class	Preferred Term	Number (%) of Subjects	Number of Events
Any Adverse Events		5 (4.1)	5
Renal and Urinary Disorders		3 (2.5)	3
	Urinary Retention	2 (1.7)	2
	Bladder Spasm	1 (0.8)	1
Gastrointestinal Disorders		2 (1.7)	2
	Abdominal Pain	1 (0.8)	1
	Rectourethral Fistula	1 (0.8)	1

**Table 4 cancers-16-02178-t004:** Adverse Events, by Preferred Term, Occurring in ≥10% of Subjects.

		Study Population (N = 121)			
		All AEs		AEs Related to the NanoKnife System	
System Organ Class	Preferred Term	Number (%) of Subjects	Number of Events	Number (%) of Subjects	Number of Events
Renal and Urinary Disorders		67 (55.4)	146	57 (47.1)	118
	Hematuria	46 (38.0)	51	38 (31.4)	43
	Dysuria	16 (13.2)	17	12 (9.9)	12
	Urinary Retention	14 (11.6)	17	13 (10.7)	16
	Micturition Urgency	14 (11.6)	15	14 (11.6)	15
	Pollakiuria	12 (9.9)	13	11 (9.1)	12
Reproductive System and Breast Disorders		31 (25.6)	46	28 (23.1)	38
	Erectile Dysfunction	12 (9.9)	12	12 (9.9)	12
	Hematospermia	12 (9.9)	12	12 (9.9)	12

**Table 5 cancers-16-02178-t005:** Adverse Events, by Preferred Term, Occurring in ≥10% of Subjects, by Timepoint.

		ITT (N = 121)					
		Ongoing at 30 Days		Ongoing at 90 Days		Overall	
System Organ Class	Preferred Term	Number (%) of Subjects	Number of Events	Number (%) of Subjects	Number of Events	Number (%) of Subjects	Number of Events
Renal and Urinary Disorders		50 (41.3)	96	30 (24.8)	51	67 (55.4)	146
	Hematuria	35 (28.9)	35	13 (10.7)	13	46 (38.0)	51
	Dysuria	13 (10.7)	13	9 (7.4)	9	16 (13.2)	17
	Urinary Retention	6 (5.0)	7	5 (4.1)	5	14 (11.6)	17
	Micturition Urgency	13 (10.7)	13	8 (6.6)	8	14 (11.6)	15
	Pollakiuria	11 (9.1)	12	6 (5.0)	7	12 (9.9)	13
Reproductive System and Breast Disorders		27 (22.3)	39	25 (20.7)	27	31 (25.6)	46
	Erectile Dysfunction	12 (9.9)	12	12 (9.9)	12	12 (9.9)	12
	Hematospermia	10 (8.3)	10	4 (3.3)	4	12 (9.9)	12

## Data Availability

The datasets presented in this article are not readily available because the data are part of an ongoing study. Data are planned to be shared upon the completion of the study and its submission to the FDA. Requests to access the datasets should be directed to ageorg@jh.edu.

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
