# Peer review of "A Description and Safety Overview of Irreversible Electroporation for Prostate Tissue Ablation in Intermediate-Risk Prostate Cancer Patients: Preliminary Results from the PRESERVE Trial"

_cancers, 2024, doi:10.3390/cancers16122178_

Round 1
Reviewer 1 Report
Comments and Suggestions for Authors
The present study is an investigational device exemption (IDE) study that allows the investigational device to be used in a clinical study to collect safety and effectiveness data. Despite the authors' reported safety data, effectiveness data, particularly the 1st year of cancer detection rate, is not provided. Instead, the authors give a PSA decline rate in 6 months, which is not a valid instrument for the oncological safety of the given device, Nanoknife. Some other points about the study are as follows:
1-Please confirm that the present report complies with the Delphi Consensus Project (Muller et al. World J Urol, 2015).
2-It should be mentioned if a central pathology was used to confirm the biopsy Gleason scores of the patients recruited at 17 centers participating in the study.
3-According to the study, the average number of electrodes to be used is 4. The authors wrote in the MM section that electrodes were placed 10-22 mm apart, which may be too much. The authors should give the mean diameter of the lesions undergoing ablation with a Nanoknife. Also, how many patients had more than one lesion to ablate?
4-Although the title of the manuscript mentions the description of the technique, the procedure of irreversible electroporation was briefly described. As far as I know, significant parts of the procedure are patient preparation, field visualization, device set-up, needle insertion, and treatment delivery. For example, in the patients with rectourethral fistula, hydro dissection was not performed. Should it be done? If not, who are the candidates?
5-There is no mention of assessing urinary and erectile dysfunction using validated questionnaires. The rate of erectile dysfunction is given in Table 5. However, we have yet to know the exact numbers with IIEF-15.
6-One patient experienced abdominal pain, and the reason for this was not given. Since this is a safety trial, please provide details.
7-The discussion section needs to be stronger. Many studies have been published on the complication rates of IRE, and they should have been mentioned in the discussion section. Please see the review by Ong et al. (Life 2021). A table might be helpful to see briefly complications more than Clavien 3.
Comments on the Quality of English LanguageThe present study is an investigational device exemption (IDE) study that allows the investigational device to be used in a clinical study to collect safety and effectiveness data. Despite the authors' reported safety data, effectiveness data, particularly the 1st year of cancer detection rate, is not provided. Instead, the authors give a PSA decline rate in 6 months, which is not a valid instrument for the oncological safety of the given device, Nanoknife. Some other points about the study are as follows:
1-Please confirm that the present report complies with the Delphi Consensus Project (Muller et al. World J Urol, 2015).
2-It should be mentioned if a central pathology was used to confirm the biopsy Gleason scores of the patients recruited at 17 centers participating in the study.
3-According to the study, the average number of electrodes to be used is 4. The authors wrote in the MM section that electrodes were placed 10-22 mm apart, which may be too much. The authors should give the mean diameter of the lesions undergoing ablation with a Nanoknife. Also, how many patients had more than one lesion to ablate?
4-Although the title of the manuscript mentions the description of the technique, the procedure of irreversible electroporation was briefly described. As far as I know, significant parts of the procedure are patient preparation, field visualization, device set-up, needle insertion, and treatment delivery. For example, in the patients with rectourethral fistula, hydro dissection was not performed. Should it be done? If not, who are the candidates?
5-There is no mention of assessing urinary and erectile dysfunction using validated questionnaires. The rate of erectile dysfunction is given in Table 5. However, we have yet to know the exact numbers with IIEF-15.
6-One patient experienced abdominal pain, and the reason for this was not given. Since this is a safety trial, please provide details.
7-The discussion section needs to be stronger. Many studies have been published on the complication rates of IRE, and they should have been mentioned in the discussion section. Please see the review by Ong et al. (Life 2021). A table might be helpful to see briefly complications more than Clavien 3.
Reviewer 2 Report
Comments and Suggestions for Authors
From a biostats and clinical epidemiology point of view, here are some comments for the Authors
- continuous covariates have to be reported only as median/IQR (mean/sd have to be omitted)
- current median follow-up for the whole cohort has to be added
- the number of progressions and deaths, if any, have to be included
- have the pts undergone a Ga68-PSMA staging PET/CT scan too? if no, why!? it's the current staging std
- figure 1, we can see point estimations, but not the interval ones! add them otherwise follow my next suggestion!
- figure 1, better to show raw PSA values rather than its modifications over time: always use raw data, if available!
- CTCAE, please add its release number
Reviewer 3 Report
Comments and Suggestions for Authors
Authors highlight the safety and efficacy of IRE prostate ablation with the NanoKnife System in patients with intermediate-risk prostate cancer (PRESERVE trial).
Preliminary results appears promising, but long term follow up should provide the final judgment for IRE in intermediate risk PC patients.
Author Response
Thank you for your review. We plan to publish on the 12-month follow up data when it is available.